# A 3-year Longitudinal Study of Pocket Money, Eating Behavior, Weight Status: The Childhood Obesity Study in China Mega-Cities

**DOI:** 10.3390/ijerph17239139

**Published:** 2020-12-07

**Authors:** Lu Ma, Zeping Fang, Liwang Gao, Yaling Zhao, Hong Xue, Ke Li, Youfa Wang

**Affiliations:** 1Global Health Institute, School of Public Health, Xi’an Jiaotong University Health Science Center, Xi’an 710000, China; maluhappy14@163.com (L.M.); fangzp@stu.xjtu.edu.cn (Z.F.); glw_0502@163.com (L.G.); yaling-zhao@163.com (Y.Z.); likekkzk@163.com (K.L.); 2Department of Health Administration and Policy, College of Health and Human Services, George Mason University, Fairfax, VA 20170, USA; hongxue0910@gmail.com

**Keywords:** family factors, pocket money, eating behaviors, overweight, obesity

## Abstract

The associations between children’s pocket money and their eating behaviors and weight status have not been examined using longitudinal data in China. Examined child and parental factors associated with children’s pocket money, and longitudinal effects of pocket money on children’s eating behaviors and weight status. Data were collected in 2015, 2016, and 2017 from 3261 school-age children and their parents in mega-cities across China (Beijing, Shanghai, Nanjing, Xi’an, Chengdu). Children’s weight, height, and waist circumference were measured; pocket money and eating behaviors were self-reported. Mixed effect models were used. Older children received more pocket money than younger children (incident rate ratio (IRR) = 1.21, 95% confidence interval (CI): 1.15, 1.26). Fathers gave their children more pocket money than mothers did (IRR = 1.22, 95% CI: 1.16, 1.30). Children with fathers having ≥ college education received more pocket money than the others did (IRR = 1.20, 95% CI: 1.04, 1.40). Some nutrition-related parenting behaviors and attitude were also associated with children’s pocket money. Compared with children receiving no weekly pocket money, those having 1–10 or 10–30 or >30-yuan weekly pocket money were 12.0–136% more likely to consume unhealthy foods and were 66–132% more likely to be overweight or obese. Some child and parental factors were associated with children’s pocket money, which increased risks of having unhealthy eating behaviors and being overweight and obese.

## 1. Introduction

Childhood obesity has become a global public health problem, including in China [1,2,3]. With the rapid development of the social economy and great changes of lifestyle, Chinese people’s eating behaviors have changed rapidly, most notably with increasing consumption of unhealthy food [4]. The prevalence of overweight and obesity (ov/ob) among Chinese children increased from 1.1% in 1985 to 18.2% in 2016, and in urban areas the rate is close to that in some developed countries [5,6].

Socioeconomic status (SES) is associated with children’s eating behaviors and weight status [7,8,9]. Pocket money is a good SES indicator of purchasing power and financial independence of children [10,11]. It gives them some autonomy in food purchasing and consumption [12]. Some cross-sectional studies, including ours, have shown that some child-, family-, and school- factors might affect pocket money children received [13,14].

With the quick economic development and increase in family income in China [15], the amount of children’s pocket money has been increasing [14]. Giving pocket money to children (especially the young ones) during holidays and special events (e.g., when children have birthdays and when some family members get married) is a popular practice in China. Some parents give monetary compensation to their children because of the parents’ increasing work-family conflicts; for example, some parents need to go to other places to work and live and must leave their children to stay with grandparents at home [16]. The use of smart phones and social media, such as WeChat, makes it much easier and common for family members to give children money. In addition, due to the “one child policy”, which started in the late 1970s and ended in 2016, many families have only one child. Family members thus tend to lavish their love through giving money to their young children [17].

At present, only very few cross-sectional studies have examined how children’s pocket money may affect their eating behaviors and weight status, including one of our recent studies [14]. Studies from European countries, Vietnam, India, and China showed that pocket money was positively associated with consumption of soft drinks, out-of-home eating, and ov/ob [14,18,19,20]. Our cross-sectional study indicated that pocket money was a risk factor for unhealthy eating behaviors and ov/ob of children from urban China. However, cross-sectional studies cannot assess causal relationships. They cannot analyze the influence of modifiable factors over time either. The associations between children’s pocket money and their eating behaviors and weight status have not been examined using longitudinal data in China.

Using longitudinal data collected from five mega-cities across China (with a population >8 million in each city), we examined: (1) changes in Chinese children’s pocket money from 2015 to 2017; (2) what parental factors were associated with children’s pocket money; and (3) how children’s pocket money might affect their eating behaviors and weight status.

## 2. Materials and Methods

### 2.1. Study Design and Participants

The Childhood Obesity Study in China Mega-cities (COCM) is an open cohort study funded by the US National Institutes of Health (NIH) aiming to examine the etiology of childhood obesity and chronic diseases in China. This study uniquely captures health trends related to lifestyle behavior changes occurring at the forefront of China’s economic growth. The COCM sampled 3365 children from 20 schools in five mega-cities across China, including Beijing (capital, North), Shanghai (Southeast), Nanjing (Southeast), Xi’an (Northwest), and Chengdu (Southwest). In each city, two primary schools and two middle schools were randomly selected, wherein one class was randomly selected from the 3rd to the 6th grades of each primary school and one from the 7th to the 9th grades of each middle school [14,21,22]. All children in the selected classes and their mothers (or another primary care giver if mothers were absent) were surveyed. Data on child growth and health, family characteristics, home environment, and energy balance-related behaviors was collected. The study data collection was approved by the Ethical Committees of the Johns Hopkins University, State University of New York at Buffalo and related collaborative institutes in China. Written informed consent was obtained from children and their parents.

Only children who participated in ≥2 surveys in the 2015, 2016, and 2017 surveys were included in this analysis. We designed the present analysis in this way to allow us to test causal relationships among the study variables. Children with missing data on age, gender, body weight, height, waist circumference, and pocket money were excluded from our analysis. The final sample size was 3261 (missing rate = 3.1%) in the present data analysis.

### 2.2. Study Variables and Measurements

#### 2.2.1. Key Study Variables

Pocket money: Children were asked to answer the following open question, “On average, how much pocket money (Chinese yuan, RMB) do you receive from your family every week?”. This variable was used both as a continuous variable and a categorical variable (0, 1–10, 11–30, and >30 RMB/week) in the data analysis.Anthropometric data: Height was measured using Seca 213 Portable Stadiometer Height-Rods (Seca China, Zhejiang, China) with a precision of 0.1 cm. Body weight was measured using Seca 877 electronic flat scales (Seca China, Zhejiang, China) with a precision of 0.1 kg. Height and weight were measured by health professionals. Child body mass index (BMI) was calculated as weight (kg) divided by the square of height (m). General overweight and obesity were defined using age- and gender-specific BMI cutoff points issued by the National Health Commission of the People’s Republic of China (underweight/normal weight: <85th percentile; 85th percentile ≤ overweight < 95th percentile; 95th percentile ≤ obesity) [23].To determine central obesity, waist circumference was measured using an inflexible tape with a precision of 0.1 cm. Waist-to-height ratio was calculated as waist circumference (cm) divided by height (cm). Central obesity was defined as having a waist-to-height ratio ≥ 0.48 and taken as a dichotomous dependent variable in mixed-effects models [24].Eating behaviors: Children were asked to report average (in days) weekly consumption of food groups during the previous 3 months. Based on the information collected, we characterized children’s eating behaviors as “unhealthy,” which included consumption of pickled food, western fast food, Chinese fast food, fried food, red meat, snacks, and beverages. We named “healthy” behaviors as consuming whole grain food, green leafy vegetables, fruits, eggs, soybeans and its products, white meat, milk, and milk products.

#### 2.2.2. Other Study Variables

We included the following factors in the models studying pocket money, weight status, and eating behaviors. These variables were included as potential predictors of the study outcomes or as variables being adjusted for in the models.

Child factors included school type (primary school, middle school), sex, age (in years), and location (Beijing, Shanghai, Nanjing, Xi’an, and Chengdu). Parental factors included paternal and maternal BMI (in kg/m^2^, with both parents’ body weight and height reported by the children’s primary caregiver), parental education (≤middle school, high and vocational schools, and ≥ college), and family home ownership (rent or share residency with relatives, own an apartment, and own a house). School factors included whether the school lacked unhealthy food restrictions.

We also investigated the associations between pocket money and nutrition-related parenting practices and attitudes: (a) nutrition-related parenting practices included weekly frequencies of maternal out-of-home eating (never, one or two times, and ≥three times) and the family’s out-of-home eating (never, one or two times, and ≥three times); (b) nutrition-related parenting attitudes: children’s parents were asked to report their normative attitudes/beliefs regarding ideal/desirable/correct nutritional practices in raising children by choosing whether they agreed or disagreed with the following statements: “child should only eat during regular meal times”, “snacks are among the best incentives for child”, “parents should not overfeed child”, “parents should be concerned about child’s future diseases due to unhealthy eating”, “parents should be concerned about child’s overweight/over-nutrition”, “parents should monitor the time and content of child’s everyday eating”, “parents should make sure child is well fed”, and “parents should encourage children to eat healthy food”. Responses to each statement were coded using a 5-point Likert scale ranging from “strongly disagree” to “strongly agree”. In the data analyses, we recoded the scale into three categories: “strongly disagree or disagree”, “does not matter”, and “strongly agree or agree”.

### 2.3. Statistical Analysis

First, descriptive statistics were calculated for the overall and gender-stratified characteristics of child weight status, eating behaviors, and pocket money in 2015–2017. Mixed-effects models were used to examine the temporal variation trend of these variables.

Second, a multilevel mixed-effects negative binomial regression model was used to examine associations between pocket money and a host of student and family characteristics, including nutrition-related parenting practices and attitudes, given skewness and the over-dispersion distribution of pocket money (in yuan). Incident rate ratio (IRR) and its 95% confidence interval (CI) was reported to indicate how much more pocket money children received on average when family factors changed by one unit.

Third, multilevel mixed-effects negative binomial regression was used to examine the longitudinal associations between pocket money and eating behaviors used as count variables displaying over-dispersion. IRR and its 95% CI were reported to indicate how much food consumption changed, on average, when the available pocket money increased by one unit. Finally, mixed-effects models were fit to test the effects of pocket money on weight status.

Gender-stratified analyses were conducted to explore potential gender differences in these associations. Analyses were conducted using STATA15.0 (Stata Corp, College Station, TX, USA). Statistical significance was set at *p* < 0.05.

## 3. Results

### 3.1. Change of Weight Status, Eating Behaviors, and Pocket Money during 2015–2017

As shown in Table 1, 69.0%, 72.4%, and 69.7% of children received pocket money in 2015, 2016, and 2017, respectively. BMI increased significantly for all children (from 19.2 to 19.4 kg/m^2^) and girls (from 18.6 to 18.8 kg/m^2^) from 2015 to 2017, but not for boys. The prevalence of central obesity increased significantly for all children (from 19.6 to 21.6%) and for boys (from 27.3 to 29.5%) from 2015 to 2017, but not for girls. No significant change was found for general ov/ob based on BMI. Consumption of healthy foods decreased significantly from 2015 to 2017, including whole grains food, fruits, eggs, soybeans and its products, while the consumption of unhealthy food increased significantly, including western fast food, Chinese fast food, and red meat.

### 3.2. Family Correlates of Child Pocket Money

Child pocket money increased with their age (IRR = 1.21, 95% CI: 1.15, 1.26). Fathers gave more pocket money to their children (IRR = 1.22, 95% CI: 1.16, 1.30) than mothers did. Children with fathers having college or higher educations received more pocket money than the others did (IRR = 1.20, 95% CI: 1.04, 1.40).

Both the frequencies of maternal (IRR = 1.03, 95% CI: 1.01, 1.06) and the family’s (IRR = 1.03, 95% CI: 1.01, 1.05) out-of-home eating were associated with more child pocket money. Children with parents who were opposed to overfeeding children received less pocket money than the others did (IRR = 0.83, 95% CI: 0.70, 0.99). Children with parents who thought that “it doesn’t matter that snacks were the best incentives for child” received more pocket money (IRR = 1.48, 95% CI: 1.09, 2.00) than the others did. Children received more pocket money if their schools lacked unhealthy food restriction (IRR = 1.34, 95% CI: 1.17, 1.53) (Table 2).

### 3.3. Longitudinal Associations between Pocket Money and Child Eating Behaviors

Pocket money was associated with the consumption of unhealthy food, with a clear dose-response relationship (Table 3). Compared with children receiving no weekly pocket money, those receiving 1–10 yuan/week, 11–30 yuan/week, and > 30 yuan/week had higher weekly frequencies of beverage consumption by 18.4%, 46.4%, and 80.6%, respectively. The patterns were similar for boys and girls.

Compared with children receiving no weekly pocket money, children receiving > 30 yuan/week consumed more whole grains (IRR = 1.06, 95% CI: 1.02, 1.11), white meat (IRR = 1.16, 95% CI: 1.10, 1.22), and milk and milk products (IRR = 1.08, 95% CI: 1.02, 1.15), while they consumed less green leafy vegetables (IRR = 0.90, 95% CI: 0.85, 0.94). Compared with children receiving no weekly pocket money, children receiving 1–10 yuan/week, 11–30 yuan/week, and > 30 yuan/week were more likely to consume western fast food, fried food, snacks, and beverages (IRR ranged from 1.12 to 2.18, *p* < 0.001). Children receiving > 30 yuan/week in pocket money were also more likely to consume pickled food (IRR = 1.27, 95% CI: 1.13, 1.44) and Chinese fast food (IRR = 1.29, 95% CI: 1.84, 2.59).

### 3.4. Longitudinal Associations between Pocket Money and Child Weight Status

Compared with children receiving no weekly pocket money, children receiving 1–10 yuan/week, 11–30 yuan/week, and > 30 yuan/week had a higher BMI by 0.16 kg/m^2^, 0.23 kg/m^2^, and 0.29 kg/m^2^, respectively. Children receiving 11–30 yuan/week and >30 yuan/week in pocket money had a 66% and a 132% higher risk of having ov/ob, respectively. Both children receiving 11–30 yuan/week and >30 yuan/week had a higher WHtR by 0.005. A similar pattern was found for girls, but not for boys (Table 4). Dose-response relationships are shown in Figure 1.

## 4. Discussion

Given the increases in children’s pocket money and obesity prevalence in China, it is of great interest to examine how pocket money, which can allow children to choose what to eat, may contribute to the epidemic of childhood obesity. Based on the three-year longitudinal data collected from five mega-cities across China (namely Beijing, Shanghai, Nanjing, Xi’an, and Chengdu), this study found that about 70% of children received some pocket money weekly during 2015–2017. Children’s age, parental education, and some of the nutrition-related parenting practices and attitude were determinants of children’s pocket money. Children receiving pocket money weekly were more likely to consume both unhealthy and healthy foods, and be overweight or obese. There are several important findings.

First, we found that on average these Chinese children received about 30 Chinese yuan (RMB) pocket money each week during 2015–2017. About 70% of children reported receiving pocket money, and >19.8% of them reported they had received >30 yuan/week. No significant changes in amounts of pocket money were found over the three years. The amount of pocket money found in our study was lower than that reported by some other studies in China [25,26,27]. The differences may be partly due to the study participants’ age. The children in our study were younger than those in the other studies (the mean age was 12 vs.16 years). Consistent with this explanation, we found that the children’s age was positively related to the amount of pocket money they received.

Indicators of rising weight status and unhealthy eating behaviors have increased from 2015 to 2017. Children’s BMI increased by 0.2 kg/m^2^ and WHtR increased by 0.002 per year on average, and the prevalence of central obesity increased by 2%. The prevalence of ov/ob and central obesity were higher in our study than that reported by other studies. For example, a study based on the Chinese National Surveys on Students’ Constitution and Health data reported that the national prevalence of general obesity and central obesity were 22.3 and 19.6%, respectively, among children in urban areas in 2014 [5,28]. These findings demonstrate that the burden of child obesity is particularly heavy in urban areas in China, especially in mega-cities. Compared with 2015, children in 2016 and 2017 had more frequent consumption of unhealthy foods such as western fast food and Chinese fast food, which may be due to the rapid economic development, global trade, and cultural exchange in China [29]. Urgent interventions are needed to reverse the epidemic of obesity and unhealthy eating behaviors in children.

Second, regarding the family correlates of pocket money, we found that children whose mothers or families frequently ate outside the home received more pocket money. This finding was consistent with our cross-sectional study in 2015 [14]. Parents supplement their care and love for their children with the provision of pocket money, which may contribute to this phenomenon [14]. Children with parents who took healthy parenting attitudes toward childcare (i.e., opposed to overfeeding child) received less pocket money, while children with parents who took unhealthy parenting attitudes (i.e., snacks are the best incentives for child) received more pocket money. Although indicators of parental attitudes associated with pocket money found in this study were not completely consistent with those in our cross-sectional study, both studies suggest that parenting attitudes were associated with the amount of pocket money children receive. Fathers gave more pocket money to children than mothers did, and children with more highly educated father received more pocket money than the others did. This is partly explained by the fact that a higher proportion of household income comes from fathers in China, and paternal education level was positively related to household income, which, in turn, influenced children’s pocket money [30].

Third, we found that pocket money was positively associated with frequencies of unhealthy food consumed (such as pickled food, western fast food, and Chinese fast food), and the IRR values increased in a gradient regardless of gender. These findings were similar to previous studies from other countries (such as Korea, Vietnam) [18,19,20,31]. Receiving more pocket money means that children have more discretionary spending power and could choose foods that they like [12]. Children tend to prefer short-term pleasures and consumption and may be influenced by incorrect information in their social environment, via commercials in the mass media, and in stores near schools, so they were more likely to choose unhealthy foods such as snacks with a wide variety of flavors and higher energy intake [18,19,20,31,32].

Our study found more pocket money (>30 yuan) increased some types of children’s healthy eating behaviors, including whole grains food, green leafy vegetables, white meat, and milk and milk product. Another study also found that pocket money could increase some types of healthy eating behaviors [19]. Having more pocket money means that children have more discretionary spending power and choose food which they like. These findings indicate that it is important to educate children about nutrition knowledge and healthy eating behaviors, and help them choose healthy food to eat. More research is needed to understand the effect of pocket money on healthy food consumption among children.

However, consumption of vegetables was negatively associated with the higher allowance. Other studies have found that vegetables were among the least preferred types of foods among children aged 4–16 years [33,34].

Fourth, we found that pocket money was a risk factor for childhood obesity. There was a positive dose-response relationship between pocket money and BMI and overweight and obesity risks. Compared with the results of our previous cross-sectional study [14], the gradient changes of βs and ORs in this study were more obvious and significant. In addition, the relationship was stronger in girls than in boys. This may be due to girls being more likely to use their pocket money to buy snacks and other food products, thus putting themselves at higher risk of overweight.

This study has several strengths. First, this longitudinal study provides novel insights on the causal relationships between pocket money, eating behaviors (including consumption of healthy and unhealthy food), and weight status (BMI, general obesity, and central obesity) in children from five mega-cities in China. We found that children’s pocket money has both favorable and unfavorable health and behavioral effects, so in order to reduce the negative impact of pocket money on children’s eating behavior and weight status and help this money play its positive role, it is important to cultivate children’s correct consumption concepts and help them develop healthy consumption behaviors. Family, school, and society need to take responsibility; specifically, parents and primary caregivers should control their children’s pocket money and guide them to develop healthy consumption behaviors. Schools should limit the supply of unhealthy food on campus; health education for children should be strengthened to reduce the consumption of unhealthy foods; and the government should restrict food advertising and marketing that target children.

This study also has some limitations. First, the data was collected from five mega-cities, thus its generalizability to other less-developed cities in China is limited. Second, some of participants were lost to follow-up during the three years. Third, child eating behaviors and parental weight and height were self-reported, so there may be potential recall bias. Fourth, our study did not measure household income variables, which may be highly correlated with child pocket money and can be a potential confounding variable. However, we used home ownership as a measure of family economic level.

## 5. Conclusions

A large proportion of children received pocket money, and some family and parental factors were predictors of the amount of pocket money children received in mega-cities in China. The pocket money could increase children’s risk of overweight and obesity, unhealthy eating behaviors, and some types of healthy eating behaviors. It is important to educate parents and other childcare providers and to empower children on how to best give and to use pocket money and help children to develop healthy eating habits. This can help fight childhood obesity in China.

## Figures and Tables

**Figure 1 ijerph-17-09139-f001:**
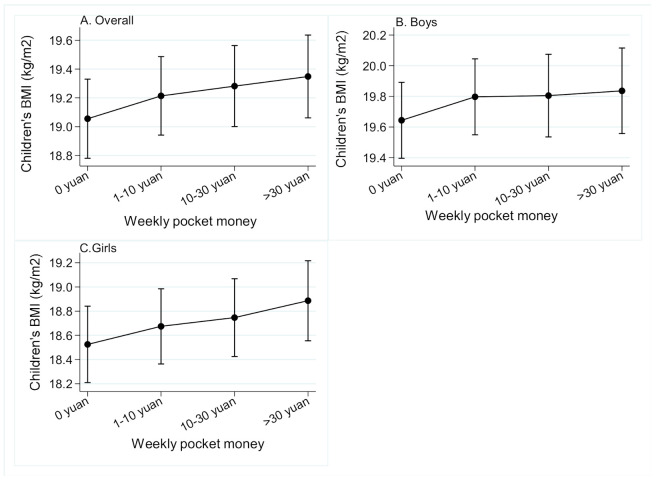
Adjusted mean of children’s BMI for each level of weekly pocket money when covariates are at their mean after mixed-effects linear regression model: The Childhood Obesity Study in China Mega-Cities. Mixed-effects model was fit to examine the longitudinal association between children’s BMI and weekly pocket money, adjusting for time, gender, age, paternal BMI, maternal BMI, paternal education, maternal education, and family homeownership. The mixed-effects model was fit for all children, boys, and girls, respectively. Location was included in the models as a random effect (intercept). The marginal means of children’s BMI was estimated based on pocket money in the mixed-effect model conditioned the other covariates was on average. As weekly pocket money changed in four categories (0, 1–10, 11–30, >30 yuan), the predicted mean of BMI was 19.0, 19.2, 19.3, 19.4 kg/m^2^ for all children, the BMI was 19.6, 19.8, 19.8, 19.8 kg/m^2^ for boys, 18.5, 18.7, 18.7, 18.9 kg/m^2^ for girls. Only children who participated in ≥ two surveys in the 2015, 2016, and 2017 surveys were included in this analysis (*n* = 3261).

**Table 1 ijerph-17-09139-t001:** Characteristics (mean/standard deviation (SD) or %) of health outcomes, weekly pocket money, and eating behaviors in children: The Childhood Obesity Study in China Mega-Cities, 2015–2017 ^a^.

Outcomes	Overall				Boys				Girls			
	2015 (*n* = 1583)	2016 (*n* = 1917)	2017 (*n* = 1935)	*p*-Value	2015 (*n* = 806)	2016 (*n* = 962)	2017 (*n* = 973)	*p*-Value	2015 (*n* = 777)	2016 (*n* = 955)	2017 (*n* = 962)	*p*-Value
1. Health Outcomes												
BMI (kg/m^2^) ^a^	19.2 (3.8)	19.3 (3.7)	19.4 (3.7)	0.002 **	19.7 (4.0)	19.7 (3.9)	20.0 (3.9)	0.173	18.6 (3.6)	18.8 (3.5)	18.8 (3.4)	0.043 *
Overweight/obesity (%) ^b^	30.8	30.2	30.6	0.203	38.3	37.4	38.2	0.221	23.0	23.0	22.9	0.819
Waist-to-height ratio ^a^	0.4 (0.1)	0.4 (0.1)	0.4 (0.1)	0.002 **	0.4 (0.1)	0.4 (0.1)	0.5 (0.1)	0.052	0.4 (0.1)	0.4(0.1)	0.4 (0.1)	0.043 *
Central obesity (%, waist-to-height ratio ≥ 0.48)	19.6	18.6	21.6	0.001 ***	27.3	26.4	29.5	0.035 *	11.6	10.7	13.5	0.392
2. Average Weekly Pocket Money												
As a continuous variable (yuan) ^c^	26.7 (64.7)	35.3 (256.8)	31.5 (111.7)	0.577	26.8 (71.7)	31.1 (88.6)	34.3 (139.0)	0.636	26.7 (56.5)	29.2 (143.1)	28.6 (74.4)	0.916
As a categorical variable (%) ^d^												
<1 yuan	31.0	27.6	30.3	0.54	32.5	29.7	32.8	0.803	29.3	25.5	27.9	0.463
1–10 yuan	28.2	33.3	29.7		26.6	31.5	29.1		30	35	30.2	
11–30 yuan	20.3	19.3	18.8		21.3	18.1	16.3		19.3	20.5	21.2	
>30 yuan	20.5	19.8	21.2		19.6	20.7	21.8		21.4	18.9	20.7	
3. Eating Behaviors (Frequency/Week)												
(1) Consumption of healthy food												
Whole grains	8.5 (5.9)	5.0 (6.8)	4.5 (5.0)	0.001 ***	8.5 (5.8)	5.3 (7.0)	4.5 (4.9)	0.001 ***	8.5 (6.1)	4.6 (6.6)	4.5 (5.0)	0.001 ***
Green leafy vegetables	6.7 (4.9)	8.4 (8.3)	7.2 (5.5)	0.314	6.4 (4.6)	8.3 (8.5)	7.1 (5.5)	0.383	7.1 (5.1)	8.4 (8.1)	7.4 (5.5)	0.642
Fruits	6.5 (4.3)	7.3 (6.7)	6.1 (4.3)	0.002 **	6.4 (4.5)	7.4 (7.1)	6.1 (4.6)	0.123	6.7 (4.1)	7.2 (6.4)	6.1 (4.0)	0.002 **
Eggs	5.2 (4.0)	5.8 (6.5)	5.0 (3.8)	0.042 *	5.2 (3.9)	6.1 (6.8)	5.2 (3.9)	0.615	5.2 (4.1)	5.5 (6.1)	4.8 (3.6)	0.015 *
Soybean and products	3.2 (3.1)	3.5 (5.1)	2.7 (2.9)	0.001 ***	3.2 (3.0)	3.6 (4.8)	2.9 (3.0)	0.018 *	3.2 (3.1)	3.4 (5.4)	2.6 (2.8)	0.001 ***
White meat	5.3 (5.2)	6.4 (7.1)	5.8 (5.9)	0.019 *	5.5 (5.0)	6.8 (7.5)	6.2 (6.1)	0.014 *	5.0 (5.3)	5.9 (6.7)	5.4 (5.7)	0.419
Milk and milk products	10.0 (6.5)	11.3 (11.4)	9.7 (6.5)	0.084	10.2 (6.6)	11.8 (12.5)	9.9 (6.9)	0.364	9.8 (6.4)	10.7 (10.3)	9.4 (6.1)	0.084
(2) Consumption of Unhealthy Food												
Pickled food	1.3 (2.1)	1.6 (2.2)	1.4 (2.0)	0.672	1.3 (2.0)	1.7 (2.1)	1.5 (2.2)	0.379	1.3 (2.1)	1.6 (2.2)	1.3 (1.7)	0.141
Western fast food	0.6 (1.0)	0.8 (1.3)	0.8 (1.3)	0.001 ***	0.7 (1.1)	0.8 (1.4)	0.9 (1.4)	0.001 ***	0.6 (0.9)	0.7 (1.3)	0.7 (1.1)	0.001 ***
Chinese fast food	1.4 (1.9)	3.3 (3.4)	2.4 (2.8)	0.001 ***	1.6 (2.0)	3.5 (3.8)	2.5 (3.0)	0.001 ***	1.2 (1.7)	3.1 (3.0)	2.3 (2.5)	0.001 **
Fried food	1.6 (1.6)	1.5 (2.0)	1.4 (1.8)	0.005 **	1.7 (1.8)	1.7 (2.0)	1.6 (2.0)	0.114	1.4 (1.4)	1.2(1.9)	1.2 (1.5)	0.012 *
Red meat	4.5 (4.0)	6.4 (6.6)	5.7 (4.9)	0.005 **	4.7 (4.0)	6.8 (6.8)	6.0 (5.1)	0.004 ***	4.3 (4.0)	6.1 (6.4)	5.3 (4.7)	0.293
Snacks	12.7 (12.7)	15.1 (16.4)	12.2 (10.2)	0.052	12.9 (14.6)	15.5 (17.8)	11.8 (10.1)	0.107	12.5 (10.4)	14.7 (14.9)	12.5 (10.3)	0.255
Beverages	9.2 (9.3)	10.4 (13.9)	8.4 (8.3)	0.003 **	10.3 (10.7)	11.6 (14.8)	9.3 (9.3)	0.024 **	8.0 (7.4)	9.2 (12.7)	7.4 (7.2)	0.038 *

Abbreviation: BMI: body mass index. Variable definition: BMI (kg/m^2^) = weight (kg)/height^2^(m). Overweight and obesity were defined using age- and gender-specific BMI cutoffs points issued by the National Health Commission of the People’s Republic of China. Waist-to-height ratio = waist circumference (cm)/height (cm). Central obesity was defined as having a waist-to-height ratio ≥ 0.48. Pocket money: each student was asked to report the average times per week that he/she ate the food groups in the last three months. ^a^. Mixed-effects linear regression model. ^b^. Multilevel mixed-effects logistic regression model. Models for weight status adjusted for city (Beijing, Shanghai, Nanjing, Xi’an and Chengdu), school (primary school, middle school), age (in years), gender, paternal BMI, maternal BMI, parental education (≤middle school, high and vocational schools, and ≥college), and family homeownership (rent or share residency with relatives, own an apartment, and own a house). ^c^. Multilevel mixed-effects negative binomial regression model. ^d^. Multilevel mixed-effects ordered logistic regression model. Models for weekly pocket money and eating behaviors adjusted for city (Beijing, Shanghai, Nanjing, Xi’an and Chengdu), school (primary school, middle school), age (in years), gender, parental education (≤middle school, high and vocational schools, and ≥college), and family living condition (rent or share residency with relatives, own an apartment, and own a house). *: *p* < 0.05; **: *p* < 0.01; ***: *p* < 0.001.

**Table 2 ijerph-17-09139-t002:** Factors associated with pocket money (incident rate ratio (IRR) and 95% CI) among children: The Childhood Obesity Study in China Mega-Cities, 2015–2017.

Child and Parental Factors	Weekly Pocket Money (in Chinese Yuan, RMB) as the Outcome Variable ^a^		
	All (*n* = 3261) IRR (95% CI)	Boys (*n* = 1642) IRR (95% CI)	Girls (*n* = 1619) IRR (95% CI)
1. Child Factors			
Time	1.03 (0.96, 1.11)	0.90 (0.78, 1.02)	1.18 (0.90, 1.53)
Gender (vs. boys)	0.96 (0.72, 1.28)	--	--
Age (years)	1.21 (1.15, 1.26) ***	1.19 (1.14, 1.24) ***	1.25 (1.19, 1.31) ***
Pocket money giver (vs. mother)			
Father	1.22 (1.16,1.30) *	1.33 (1.13, 1.59) **	1.17 (1.01, 1.35) *
Grandparent	1.02 (0.73, 1.42)	0.85 (0.54, 1.35)	1.18 (0.84, 1.66)
Student residence (vs. family)			
School	1.01 (0.67, 1.54)	1.22 (0.76, 1.98)	0.92 (0.51, 1.68)
2. Family SES			
Paternal highest education (vs. ≤middle school)			
High and vocational schools	1.27 (0.90, 1.80)	1.34 (1.08, 1.67) **	1.22 (0.78, 1.88)
≥College	1.20 (1.04, 1.40) *	1.56 (1.27, 1.91) ***	0.94 (0.71, 1.26)
Maternal highest education (vs. ≤middle school)			
High and vocational schools	0.85 (0.69, 1.06)	0.79 (0.66, 0.93) **	0.96 (0.71, 1.30)
≥College	0.82 (0.67, 1.06)	0.67 (0.55, 0.80) ***	1.07 (0.67, 1.71)
Family homeownership (vs. rent or share residency with relatives)			
Own an apartment	1.02 (0.88, 1.18)	0.96 (0.68, 1.36)	1.08 (1.03, 1.14) **
Own a house	1.02 (0.88, 1.17)	0.73 (0.55, 0.96) *	1.29 (1.07, 1.56) **
3. Nutrition-Related Parenting Behavior			
The weekly frequency of family’s out-of-home eating	1.03 (1.01, 1.05) **	1.06 (1.03, 1.08) ***	1.00 (0.95, 1.04)
The weekly frequency of mother’s out-of-home eating	1.03 (1.01, 1.06) *	1.02 (0.99, 1.05)	1.05 (1.03, 1.08) ***
Whether family often eats dinners together	1.07 (0.91, 1.26)	1.14 (1.05, 1.24) **	1.01 (0.78, 1.32)
4. Nutrition-Related Parenting Attitude			
Snacks were among the best incentives for child (vs. Disagree or strongly disagree)			
Not matter	1.48 (1.09, 2.00) *	1.92 (1.38, 2.66) ***	1.22 (0.96, 1.55)
Agree or strongly agree	1.14 (0.94, 1.40)	1.33 (0.85, 2.08)	0.91 (0.76, 1.09)
Parents should not overfeed child (vs. Disagree or strongly disagree)			
Not matter	0.86 (0.75, 0.99) *	0.82 (0.57, 1.18)	0.99 (0.70, 1.39)
Agree or strongly agree	0.83 (0.70, 0.99) *	0.89 (0.60, 1.31)	0.76 (0.58, 0.99) *
5. School factors			
Whether school had unhealthy food restriction (vs. Yes)			
No	1.34 (1.17, 1.53) ***	1.45 (1.00, 2.12)	1.21 (0.87, 1.70)

Abbreviations: IRR: incident rate ratio; 95% CI: 95% confidence interval. ^a^. Pocket money: each student was asked to report “on average, how much pocket money (RMB) do you receive from your family every week?”, and this variable was used as a continuous variable in this data analysis. We used multilevel mixed-effects negative binomial regression model to analyze factors associated with pocket money, and the independent variables included child factors (gender, age, pocket money giver, and student residence), family social-economic status (parental education, family homeownership), parental nutrition-related parenting behavior (weekly frequency of family’s out-of-home eating and mother’s out-of-home eating, whether family often eats dinners together), parental nutrition-related parenting attitudes (snacks are among the best incentives for child, parents should not overfeed child), and school factors (whether school have unhealthy food restriction). Parenting attitudes on the following statements: “child should only eat during regular meal times”, “parents should be concerned about child’s future diseases due to unhealthy eating”, “parents should be concerned about child’s overweight/overnutrition”, “parents should monitor the time and content of child’s everyday eating”, “parents should make sure your child is well fed”, “parents should encourage your children to eat healthy food” were also adjusted in the model. Associations between these variables and pocket money were not statistically significant and thus were not presented in the table due to space limit. *: *p* < 0.05; **: *p* < 0.01; ***: *p* < 0.001.

**Table 3 ijerph-17-09139-t003:** Longitudinal associations (IRR and 95% CI) between children’s pocket money and their eating behaviors: The Childhood Obesity Study in China Mega-Cities, 2015–2017.

Eating Behaviors, ^a^ (Times/Week)	IRR (95%CI) (“<1 Chinese Yuan” Was Used as the Reference Group for the Other Three Categories of Weekly Pocket Money)								
	Overall (*n* = 3261)			Boys (*n* = 1642)			Girls (*n* = 1619)		
	“1–10 yuan”	“10–30 yuan”	“>30 yuan”	“1–10 yuan”	“10–30 yuan”	“>30 yuan”	“1–10 yuan”	“10–30 yuan”	“>30 yuan”
Consumption of healthy food									
Whole grains food	1.01 (0.98, 1.05)	1.02 (0.97, 1.08)	1.06 ** (1.02, 1.11)	0.96 (0.88, 1.04)	0.98 (0.87, 1.11)	1.09 (0.96, 1.25)	1.04 (0.91, 1.19)	1.03 (0.88, 1.20)	1.01 (0.83, 1.23)
Green leafy vegetables	0.93 (0.85, 1.02)	0.92 * (0.86, 0.99)	0.90 *** (0.85, 0.94)	0.91 (0.80, 1.02)	0.94 (0.85, 1.04)	0.91 (0.82, 1.01)	0.95 (0.88, 1.03)	0.91 (0.78, 1.06)	0.88 (0.76, 1.02)
Fruits	0.98 (0.93, 1.03)	0.98 (0.90, 1.06)	1.02 (0.98, 1.07)	0.98 (0.90, 1.06)	0.96 (0.88, 1.06)	1.02 (0.94, 1.11)	0.99 (0.93, 1.05)	0.99 (0.89, 1.09)	1.02 (0.94, 1.10)
Eggs	1.01 (0.92, 1.10)	0.96 (0.89, 1.04)	0.99 (0.95, 1.03)	0.97 (0.89, 1.06)	0.93 (0.85, 1.02)	0.98 (0.93, 1.04)	1.03 (0.90, 1.18)	0.99 (0.88, 1.12)	0.99 (0.89, 1.12)
Soybean and products	1.04 (1.00, 1.11)	1.05 (0.93, 1.19)	1.08 (0.96, 1.23)	1.04 (0.98, 1.10)	0.97 (0.86, 1.10)	1.05 (0.91, 1.22)	1.05 (0.91, 1.20)	1.15 (0.96, 1.38)	1.12 (0.96, 1.32)
White meat	1.04 (0.95, 1.14)	1.05 (0.99, 1.11)	1.16 *** (1.10, 1.22)	1.01 (0.94, 1.10)	1.01 (0.92, 1.10)	1.18 * (1.12, 1.24)	1.06 (0.96, 1.17)	1.08 (0.98, 1.18)	1.12 * (1.01, 1.24)
Milk and milk products	1.04 (1.00, 1.10)	1.04 (0.98, 1.10)	1.08 * (1.02, 1.15)	1.03 (0.96, 1.11)	1.03 (0.94, 1.12)	1.09 * (1.01, 1.19)	1.06 (0.99, 1.14)	1.04 (0.96, 1.14)	1.08 (0.99, 1.18)
Consumption of unhealthy food									
Pickled food	1.09 (0.99, 1.21)	1.25 *** (1.12, 1.40)	1.27 *** (1.13, 1.44)	1.18 (0.95, 1.47)	1.43 *** (1.20, 1.69)	1.30 *** (1.16, 1.46)	1.01 (0.87, 1.18)	1.10 (0.99, 1.23)	1.23 (0.93, 1.61)
Western fast food	1.20 *** (1.12, 1.28)	1.57 *** (1.34, 1.84)	2.18 *** (1.84, 2.59)	1.20 * (1.04, 1.39)	1.61 *** (1.32, 1.96)	2.36 *** (1.91, 2.92)	1.16 ** (1.04, 1.30)	1.50 *** (1.23, 1.84)	2.00 *** (1.66, 2.42)
Chinese fast food	0.96 (0.84, 1.10)	1.12 (0.95, 1.33)	1.29 ** (1.09, 1.52)	0.92 (0.75, 1.14)	1.08 (0.85, 1.37)	1.20 (1.00, 1.45)	1.01 (0.88, 1.15)	1.17 * (1.01, 1.35)	1.39 * (1.08, 1.80)
Fried food	1.19 *** (1.09, 1.31)	1.30 *** (1.21, 1.39)	1.56 *** (1.35, 1.82)	1.17 (0.99, 1.38)	1.29 ** (1.10, 1.51)	1.59 *** (1.32, 1.91)	1.21 * (1.02, 1.44)	1.30 *** (1.19, 1.43)	1.52 *** (1.21, 1.92)
Red meat	1.00 (0.88, 1.12)	1.02 (0.92, 1.13)	1.05 (0.98, 1.13)	0.95 (0.87, 1.04)	1.01 (0.93, 1.10)	1.12 *** (1.06, 1.18)	1.05 (0.90, 1.21)	1.02 (0.86, 1.20)	0.98 (0.86, 1.11)
Snacks	1.12 ** (1.04, 1.22)	1.29 *** (1.21, 1.37)	1.42 *** (1.33, 1.53)	1.12 (0.97, 1.28)	1.30 ** (1.11, 1.53)	1.43 *** (1.20, 1.70)	1.14 * (1.02, 1.28)	1.28 *** (1.20, 1.36)	1.41 *** (1.30, 1.52)
Beverages	1.18 *** (1.10, 1.28)	1.46 *** (1.38, 1.56)	1.81 *** (1.56, 2.10)	1.19 ** (1.08, 1.32)	1.52 *** (1.34, 1.73)	1.89 *** (1.57, 2.28)	1.18 ** (1.07, 1.30)	1.41 *** (1.31, 1.52)	1.71 *** (1.49, 1.96)

Abbreviations: IRR, incident rate ratio; 95% CI, 95% confidence interval. ^a^. Pocket money: each student was asked to report “on average, how much pocket money (RMB) do you receive from your family every week?”. Multilevel mixed-effects negative binomial regression model was used to analyze the longitudinal associations between pocket money and children’s eating behaviors. All models adjusted for time, city (Beijing, Shanghai, Nanjing, Xi’an and Chengdu), gender, age (in years), paternal BMI, maternal BMI, paternal education (≤middle school, high and vocational schools, and ≥college), maternal education (≤middle school, high and vocational schools, and ≥college), family homeownership (rent or share residency with relatives, own an apartment, and own a house), weekly frequency of family’s out-of-home eating and mother’s out-of-home eating, and parental nutrition-related parenting attitudes (“child should only eat during regular meal times”, “snacks are among the best incentives for child”, “parents should not overfeed child”, “parents should be concerned about child’s future diseases due to unhealthy eating”, “parents should be concerned about child’s overweight/overnutrition”, “parents should monitor the time and content of child’s everyday eating”, “parents should make sure your child is well fed”, and “parents should encourage your children to eat healthy food”). *: *p* < 0.05; **: *p* < 0.01; ***: *p* < 0.001.

**Table 4 ijerph-17-09139-t004:** Longitudinal associations between children’s pocket money and their body weight status: The Childhood Obesity Study in China Mega-Cities, 2015–2017.

Pocket Money	BMI (kg/m^2^) ^a^ Beta (95%CI)	Overweight/Obesity ^b^ OR (95%CI)	WHtR ^a^ Beta (95%CI)	Central Obesity ^b^ OR (95%CI)
All (*n* = 3261)				
Weekly pocket money (vs. 0 yuan)				
1–10 yuan	0.16 (0.03, 0.29) *	1.51 (1.00, 2.30)	0.002 (−0.001, 0.005)	1.08 (0.74, 1.57)
11–30 yuan	0.23 (0.06, 0.39) **	1.66 (1.01, 2.74) *	0.005 (0.001, 0.008) **	1.42 (0.90, 2.24)
>30 yuans	0.29 (0.12, 0.47) ***	2.32 (1.38, 3.91) **	0.005 (0.001, 0.008) *	1.50 (0.94, 2.40)
Boys (*n* = 1642)				
Weekly pocket money (vs. 0 yuan)				
1–10 yuan	0.15 (−0.04, 0.35)	1.13 (0.63, 2.02)	0.001 (−0.004, 0.006)	0.85 (0.48, 1.50)
11–30 yuan	0.16 (−0.08, 0.40)	1.62 (0.80, 3.25)	0.005 (−0.001, 0.011)	1.41 (0.72, 2.77)
>30 yuans	0.19 (−0.06, 0.45)	1.79 (0.88, 3.67)	0.002 (−0.004, 0.008)	1.18 (0.59, 2.36)
Girls (*n* = 1619)				
Weekly pocket money (vs. 0 yuan)				
1–10 yuan	0.15 (−0.02, 0.32)	2.29 (1.18, 4.47) *	0.004 (0.001, 0.007) *	2.43 (1.22, 4.84) *
11–30 yuan	0.22 (0.02, 0.42) *	2.02 (0.92, 4.41)	0.005 (0.001, 0.009) *	1.67 (0.75, 3.74)
>30 yuans	0.36 (0.14, 0.58) ***	3.36 (1.46, 7.72) **	0.008 (0.004, 0.012) ***	2.75 (1.20, 6.28) *

Abbreviations: OR: odds ratio; 95%CI: 95% confidence interval; BMI: body mass index; WHtR: waist-to-height ratio. Variable definition: BMI (kg/m^2^) = weight (kg)/height^2^ (m). Waist-to-height ratio = waist circumference (cm)/height (cm). Overweight and obesity were defined using age- and gender-specific BMI cutoffs points issued by the National Health Commission of the People’s Republic of China. Central obesity was defined as having a waist-to-height ratio ≥ 0.48. All models adjusted for time, gender, age (in years), paternal BMI, maternal BMI, paternal education (≤middle school, high and vocational schools, and ≥college), maternal education (≤middle school, high and vocational schools, and ≥college), and family homeownership (rent or share residency with relatives, own an apartment, and own a house). ^a^. Mixed-effects linear regression model was used. ^b^. Multilevel mixed-effects logistic regression model was used. *: *p* < 0.05; **: *p* < 0.01; ***: *p* < 0.001. Numbers in bold indicated statistical significance, *p* < 0.05.

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
