# Peer review of "A 3-year Longitudinal Study of Pocket Money, Eating Behavior, Weight Status: The Childhood Obesity Study in China Mega-Cities"

_ijerph, 2020, doi:10.3390/ijerph17239139_

Round 1

Reviewer 1 Report

This is an interesting longitudinal work, well thought out and with a good statistical analysis. There are only a few minor issues that should be reviewed.
-In the abstract the background that justifies the need to carry out the work and its innovative aspect are absent. I would recommend that you include a couple of lines about this.
-The citation of references throughout the text is sometimes pasted at the end of the sentence and in others after a space, please standardize throughout the manuscript.
-In children, a change in weight and height over time is normal. The authors propose an increase in the BMI but there is no discussion of how much an "abnormal" increase would be.
-On line 63 and throughout the text in the BMI formula there is an error. The units are m and not m2 as the authors state. I understand that the correct expression would be (height2) (m). Throughout the text the m2 is not written with superscript.
-In point 3.4 the term "influence of pocket money on weight" infers causality and perhaps is too blunt. Given the number of environmental and socioeconomic factors that influence the development of obesity, I would recommend that you moderate the expression. Regardless of how much money children receive, it cannot be ruled out that belonging to a family from a lower economic stratum or low education may be a factor in obesity.
-Figure 1 can remove the expression "in 4 categories".
-References 5,16,30 are complete?
-Table name and figure are in bold except in Table 4.

Author Response

Response to Reviewer 1 Comments

This is an interesting longitudinal work, well thought out and with a good statistical analysis. There are only a few minor issues that should be reviewed. 

Point 1: In the abstract the background that justifies the need to carry out the work and its innovative aspect are absent. I would recommend that you include a couple of lines about this. 

Response 1: Done.

Point 2: The citation of references throughout the text is sometimes pasted at the end of the sentence and in others after a space, please standardize throughout the manuscript.

Response 2: All the citations of references throughout the text were put after a space.   

Point 3: In children, a change in weight and height over time is normal. The authors propose an increase in the BMI but there is no discussion of how much an "abnormal" increase would be. 

Response 3: In Table 1, the BMI significantly increased from 2015 to 2017 among all children and girls. We used “the prevalence of overweight and obesity” to indicate the “abnormal” increase, and found that the prevalence of overweight and obesity among all children, boys, and girls did not increase significantly during 2015 and 2017. 

Point 4: On line 63 and throughout the text in the BMI formula there is an error. The units are m and not m2 as the authors state. I understand that the correct expression would be (height2) (m). Throughout the text the m2 is not written with superscript. 

Response 4: We changed the unit of height to m, and revised the expression throughout the text.  

Point 5: In point 3.4 the term "influence of pocket money on weight" infers causality and perhaps is too blunt. Given the number of environmental and socioeconomic factors that influence the development of obesity, I would recommend that you moderate the expression. Regardless of how much money children receive, it cannot be ruled out that belonging to a family from a lower economic stratum or low education may be a factor in obesity. 

Response 5: We revised “influence of pocket money on” to “longitudinal associations between pocket money and weight status” and “longitudinal associations between pocket money and eating behaviors”. In the longitudinal data analyses, age, gender, paternal and maternal BMI, paternal and maternal education, and family homeownership were adjusted as covariates.  

Point 6: Figure 1 can remove the expression "in 4 categories". 

Response 6: Done.

Point 7: References 5,16,30 are complete?

Response 7: Yes, they are complete.

Point 8: Table name and figure are in bold except in Table 4.

Response 8: All the names of tables and figure were changed to not in bold.

Reviewer 2 Report

I found the article interesting and I have only some concrete suggestions/comments for the authors' considerations. All included below:

Lines 89-90 It would be useful to have an idea about what could be purchased with the money included in the first two categories as that could affect what the child buys

Line 91. Suggest using the term Anthropometric data as it includes weight, height, waist circumference

Lines 95-96 “General overweight and obesity were defined using age- and gender-specific BMI cutoff points issued by the National Health Commission of the People's Republic of China [23]” Would it be possible to indicate how the criteria relates with the WHO cut-off points? As a reference?

Line 112 and table 2 Could the authors indicate how they divide the children according to age? (This relates with table 2, where age is presented as a factor that relates with risk) Is there a cut-off point? Is it that the elder the child the more risk of unhealthy behaviour?

Line 114 How reliable is to have height/weight of parents reported by primary caregiver? Was there a way to check reliability of that information?

Lines 153-154 & Table 1 The authors indicate that there was an increase in BMI between 2015 and 2017 (in general and for girls) Looking at the table there is a decrease in percentage of overweight/obese children (numbers in parenthesis Is that correct? If so, could it be commented? If not, could the authors review the way data is presented in the table?

Table 3. Is interesting to find that the higher the amount of pocket money the more consumption of whole grain foods (a healthy food) It might be good to comment on this point. The same goes for the consumption of white meat (that is considered better option than red meat)

Line higher If there is an increase of WHtR by 0.005 and 0.005 for both 11-30 and >30 yuan I suggest giving the number only once.

Lines 239-240 239 It indicates that WHtR is weight to height ratio.; please make the correction as it is waist circumference to height ratio

Author Response

Response to Reviewer 2 Comments

I found the article interesting and I have only some concrete suggestions/comments for the authors' considerations. All included below:

Point 1: Lines 89-90 It would be useful to have an idea about what could be purchased with the money included in the first two categories as that could affect what the child buys.

Response 1: Thanks for the useful comments. However, no such data were collected in this study.

Point 2: Line 91. Suggest using the term Anthropometric data as it includes weight, height, waist circumference.

Response2: Done.

Point 3: Lines 95-96 “General overweight and obesity were defined using age- and gender-specific BMI cutoff points issued by the National Health Commission of the People's Republic of China [23]” Would it be possible to indicate how the criteria relates with the WHO cut-off points? As a reference?

Response 3: The cut-off points for the Chinese standard were “Underweight/normal weight: < 85th percentile; 85th percentile ≤ overweight < 95th percentile; 95th percentile ≤ Obesity”. It is the same percentile cut-off as the standard of the US, but it is not related to the WHO cut-off points. The related information was added in the manuscript.  

Point 4: Line 112 and table 2 Could the authors indicate how they divide the children according to age? (This relates with table 2, where age is presented as a factor that relates with risk) Is there a cut-off point? Is it that the elder the child the more risk of unhealthy behaviour?

Response 4: Age was used as a continuous variable in this study, thus, no cut-off points were used. Age was adjusted as a covariate when analyzed the associations between pocket money and eating behaviors. We found that age was positively associated with unhealthy eating behaviors among all children (β=0.55, 95%CI: 0.47, 0.63), boys (β=0.53, 95%CI:0.40,0.65), and girls (β=0.60, 95%CI:0.50,0.70).  

Point 5: Line 114 How reliable is to have height/weight of parents reported by primary caregiver? Was there a way to check reliability of that information?

Response 5: The height/weight of parents reported by primary caregiver were relatively reliable in this study. More than 60% of children’s primary caregivers were their mothers or fathers (71.7%, 70.7%, and 58.2% in 2015, 2016, and 2017, respectively). 97.6% of the parent questionnaire was reported by children’s fathers or mothers. Therefore, most of the height and weight were self-reported or partner-reported by children’s parents. However, we did not measure parental height or weight, we added this as a limitation.    

Point 6: Lines 153-154 & Table 1 The authors indicate that there was an increase in BMI between 2015 and 2017 (in general and for girls) Looking at the table there is a decrease in percentage of overweight/obese children (numbers in parenthesis Is that correct? If so, could it be commented? If not, could the authors review the way data is presented in the table?

Response 6: In Table 1, there was only small significant increase in BMI between 2015 and 2017 in all children and girls. No significant increase in the prevalence of overweight and obesity were found between 2015 and 2017 among all children, boys, and girls.   

Point 7: Table 3. Is interesting to find that the higher the amount of pocket money the more consumption of whole grain foods (a healthy food) It might be good to comment on this point. The same goes for the consumption of white meat (that is considered better option than red meat)

Response 7: We added the comments on the “positive associations between pocket money and consumption of healthy foods ”in the Discussion part.

Point 8: Line higher If there is an increase of WHtR by 0.005 and 0.005 for both 11-30 and >30 yuan I suggest giving the number only once.

Response 8: The sentence was changed to “Both children receiving 11-30 yuan/week and > 30 yuan/week had a higher WHtR by 0.005”.

Point 9: Lines 239-240 239 It indicates that WHtR is weight to height ratio.; please make the correction as it is waist circumference to height ratio.

Response 9: Done.